**Understanding the Role of 2019 Amazon Wildfires on Antarctic Sea Ice Extent Using Data Science Approaches**

Sudip Chakraborty[1], Chhaya Kulkarni[1], Atefeh Jabeli[1], Akila Sampath[1], Gehan Boteju[1], Jianwu Wang[1], and Vandana Janeja[1].

1. Institute for Harnessing Data and Model Revolution in the Polar Regions (iHARP), the University of Maryland, Baltimore County, MD, USA.

**ABSTRACT:**

This study investigates the impacts of the black carbon (BC) aerosols generated during the 2019 extreme Amazon wildfire events on the sea ice extent over the Antarctic region. Random forest (RF), elastic net regression (EL), matrix profile (MPF), and causal discovery (CD) analysis have been employed on a suite of satellite measurements for this analysis. In 2019, a higher number of BC aerosol atmospheric rivers (AAR) that transport the aerosols from the Amazon region arrived in the Antarctic region compared to 2018. It has been observed that between August 2019 and February 2020, SIE loss over the Antarctic region increased threefold (487 Gt) than the mean SIE loss (143 Gt per year) from 2002. The Weddell, Ross Sea (Ross), and Indian Ocean (IO) regions experienced higher loss in SIE during BC AAR days during that period, with the Weddell region topping the chart. Bell-Amundsen (BA) and the Pacific Ocean (PC) region were the least affected and showed the minimum and insignificantly different SIE loss as compared to the previous year. Our analysis shows that the ice surface over the Antarctic peninsula was darker in 2019. RF, CD, and EL show that the shortwave upward radiative flux or the reflected sunlight, temperature, longwave upward energy, or the emitted radiation from the earth are the most important factors that influence the SIE loss over the Weddell, Ross, and Indian Ocean regions. RF and EL analyses were unable to capture the influence of wind and precipitation on SIE over BA; however, CD analysis captures the relationship. MPF shows that the highest (lowest) number of discords were found over the Weddell (BA) region - confirming the largest (lowest) loss in SIE over there. MPF also finds a higher number of discords in SIE occurring over the Ross region between August 2019 and February 2020 than the previous year, thus can explain the higher SIE loss during the presence of BC AARs over there.

**Abbreviations:** Shortwave downward (SWD); shortwave upward (SWU); longwave upward (LWU); and longwave downward (LWD) at the surface during clear sky, Precipitation (PPT), Temperature (T), specific humidity (Q), Black carbon (BC), Aerosol Atmospheric River (AAR), Sea ice extent (SIE), Aerosol optical depth (AOD), Zonal wind (U) and meridional wind (V), Indian Ocean (IO), Bell-Amundsen (BA), the Pacific Ocean (PC), Random forest (RF), elastic net regression (EL), matrix profile (MPF), and causal discovery (CD).

 **KEYWORDS:** Antarctic Ice Sheet Melt, wildfire Aerosols, Amazon wildfire, Albedo.

**ACM Reference Format:** Anonymous Author(s). 2023. Understanding the Role of 2019 Amazon Wildfires on Antarctic Sea Ice Extent Using Data Science Approaches. In Proceedings of KDD'23 Fragile Earth Workshop. ACM, Long Beach, CA, USA, 8 pages. https://doi.org/XXXXXXX XXXXXX

## 1. INTRODUCTION

The recent and unprecedented rise in sea level is posing great threats to human civilization with about 60% of the global population living within 100 km of a sea or oceanic coast [1]. With the advent of the satellite era along with the coastal tide gauges it has been observed that the global mean sea level has risen by 10 cm since 1993 and is projected to increase between 26–98 cm of sea level rise in the future[2]. The sea level rise is primarily caused by the melting of the land ice and sea ice sheets in Greenland and Antarctica[3]. Ice sheets in the Antarctic region are losing ice mass due to melting at an average rate of 150 Gt per year[4]. Sea ice creates a barrier that separates

the ocean from the atmosphere. In addition to keeping sunlight out, sea ice traps existing heat in the ocean, keeping it from warming the air above. This ability of the ice to keep heat in the ocean depends on its extent and its thickness[5]. Accelerated melting of the sea ice will eventually expose the land ice to warm water and the associated melting of the land ice will contribute to an extreme sea level rise in the future. Thus, for a clear understanding and prediction of the future sea level rise, understanding sea ice melting is crucial for the sustainability of society and the ecosystem surrounding us.

When sea ice melts, darker-colored liquid water is left exposed to absorb sunlight. As the snow albedo (a measure of light reflection) decreases, the amount of reflected sunlight to the space decreases and is rather absorbed by the land surface making it warmer and leading to melt[6]. Warmer water and the land surface accelerate the melting of ice, creating a positive ice-albedo feedback cycle. Snow darkening by black carbon (BC) aerosols significantly amplifies the greenhouse effect by two times[6] and accelerates ice sheet melting[7] because they absorb sunlight, warm the surface where they deposit, and darken the snow and ice surface[8]. Aerosol atmospheric rivers (AAR), long and elongated channels of strong wind and extreme mass transport, can transport these aerosols to long distances - often intercontinental. Studies show that BC particles generated over the US and East Asia can often reach Greenland due to AAR activities[9], [10]. Only 20-30 of such activities in a year can transport 40-80% of the total annual transport of BC particles in Greenland. The snow darkening phenomena are very complex and how BC aerosols affect the snow albedo, modulate the radiative properties like surface temperature and radiation, and accelerate the melting process is still unknown[9], [10].

In 2019, the Amazon rainforest witnessed the worst-ever deforestation and man-made wildfire events[11] for agricultural purposes[12]. The wildfires generated many BC AARs that arrived over the Antarctic region. Based on the satellite measurements from various NASA satellites[13], the average loss of sea ice over the Antarctic region is estimated to be 148 Gt per year since 2002. However, the sea ice melt increased threefold during the melting period (Austral spring and summer). The estimated loss of sea ice was 487 Gt (from 2288 Gt loss on August 15th, 2019 to 2775 Gt loss on February 14th, 2020 since 2002) over the Antarctic region[13]. Owing to such an acceleration of the sea ice melting, it is of primary importance to investigate if the Antarctic Sea ice is vulnerable to forest fire-generated aerosols from the Amazon. If the aerosol deposition over the sea ice has exacerbated the ongoing sea ice melting due to global warming, it is of utmost importance for the climate community to focus on the imminent threat of Amazon wildfires not only on the sea ice but also on the land ice melting over the Antarctic region.

This paper takes advantage of the multiyear remote sensing measurements from various satellites (Table 1) over the Antarctic region (60°S-90°S; 180°W to 180°E). We use various measurements of radiative properties, precipitation, humidity, and AAR, aerosol optical depth to infer the plausible reasons for the extreme sea ice melt. In this study, we estimate the change in the sea ice extent (SIE) over the Antarctic. The sea ice extent data has been obtained from the National Snow and Ice Data Center. The sea ice extent data has been obtained from various sensors (Table 1) and has been extensively used to study sea ice changes[14]–[16]. We employ various machine learning methods to understand the role of aerosols on the exacerbated sea ice loss in terms of their extent in $km^2$, such as Random Forest (RF), Elastic Net (EN), and Matrix profile analyses that are described in detail in the methodology section. The choice of machine learning techniques has been adopted based on their usefulness for this particular kind of study and the previous studies that have successfully implemented these techniques to study and address various climate change-related problems[17]–[21].

## 2. DATASETS:

| Satellite | parameters | Resolution | Unit |
|---|---|---|---|
| CERES | SWD, SWU, LWD, LWU | 1° | w/m$^2$ |
| GPM IMERG | PPT | 1° | mm |
| AIRS | T and Q | 1° | K; kg/kg |
| AAR data | BC AAR | 0.5° x 0.625° | |
| SMMR, SSM/I, SSMIS and NIMBUS | SIE | daily averaged | km$^2$ of SIE in a grid |
| MERRAero/MODIS | AOD | 0.5° x 0.625° | No unit |
| ERA Interim | U and V | 0.25° | m/s |

**Table 1:** Name of the satellites and list of the parameters used with acronyms and units. All the datasets are between 2018-2020 and have been preprocessed at 0.25° resolution at daily scale.

## 3. METHODOLOGY

### 3.1 Data preprocessing and domain

Owing to the different resolutions of the datasets used in this study (Table 1), we used xESMF Python API to convert all the datasets to 0.25°. After processing the data, thorough and careful checks have been performed to assess if any loss has occurred during processing the data from different resolutions to 0.25° over different regions in Antarctica by comparing the datasets between their native resolution and the processed data. Table 2 summarizes the longitudinal range for all five regions namely the Ross Sea (Ross), the Weddell region (Weddell), the Indian Ocean (IO), the Bell-Amundsen (BA) region, and the Pacific Ocean (PC). Longitudinal interval is assigned to each of the five different polar regions of the Antarctic basin. An N × P matrix has been created for each region where N represents the number of days with no missing values present in each data and P denotes the number of features. Similarly, we have created a matrix of the SIE parameter containing data from N number of days.

It is important to note that the sea ice melt is stronger during the Austral spring and summer. An example is included in Figure 2 showing the SIE between 2018 and 2020 over the Weddell region is prone to sea ice loss. Thus, we focus on the SIE loss during August 2018 - February 2019 (season18) and August 2019- February 2020 (season19) to assess the impact of the BC aerosols transported from the Amazon region. We calculate the differences in the number of AARs arrived, SIE loss, and number of discords in SIE over the region between the two-time frames mentioned above. We also show our interpretation of the possible causes of the SIE loss during 2018-2020 by using RF, EN, and CD analysis.

### 3.2 Matrix profile

Matrix profile (MPF)[22], [23] is a data structure that uses similarity search algorithms to identify patterns such as discords - unusual patterns and motifs - frequently occurring patterns. Time series discords have emerged as an efficient and competitive anomaly detection method. Time series discords refer to subsequences of a time series that are most unusual: those that are maximally distinct from all other subsequences in the same time series. Time series discords are primarily used to identify anomalies in long-term time series. MPF is useful in determining the changes in terms of unusualness in both the locations and the time periods. We employed discord detection

in this study to determine the regions exhibiting a higher number of discords in the five Antarctic regions.

| Antarctic Region | Longitudinal Interval | # discords Season18 | # discords season19 |
|---|---|---|---|
| Indian Ocean | 20° to 90° | 26 | 23 |
| Pacific Ocean | 90° to 160° | 20 | 12 |
| Bel/Amundsen Sea | 230° to 300° | 10 | 11 |
| Ross Sea | 160° to 230° | 26 | 31 |
| Weddell Sea | 300° to 20° | 43 | 25 |

**Table 2:** Longitudinal range for five regions and the number of Discords in different Antarctic Regions.

Discords are computed by sliding a fixed subsequence (a small portion of time series) across the multiple subsequences. A matrix profile is a vector storing the (z-normalized) Euclidean distance (the straight-line distance between two points) between any subsequence in a time series and its nearest neighbor. A discord is with respect to a specific attribute $a_j$ that holds values that are distinctly different from those of its neighbors in the dataset[24], [25]. This phase emphasizes the need to devise methods for determining the size of the window.

### 3.3 Causal discovery

This study used the PCMCI (Partial Correlation-based Causal Discovery Algorithm)[26] as the primary analysis tool to explore the causal relationships among various features in five different polar regions of the Antarctic basin. The PCMCI algorithm is renowned for its efficacy in capturing causal dependencies within multivariate time series data. To determine the significance of the identified causal relationships, a p-value threshold of 0.05 was applied. Additionally, the investigation incorporated time lags of 1, 2, and 3 to account for potential lagged effects. The experimental design was meticulously devised to unravel the intricate causal interactions among the examined features and the parameter of interest, namely the sea ice extent. By leveraging the PCMCI algorithm, the study aimed to elucidate the underlying causal dynamics governing the relationships between the features and the sea ice extent.

### 4. RESULTS

Figure 1A shows the differences in the number of AARs arriving over the Antarctic region between season18 and season19. It appears from Fig. 1A that over the BA, Weddle, and Ross Sea region the numbers of BC AARs arriving during season19 than the previous season are higher (5-10 AARs) with the Ross region experiencing more than 10 AARs than the previous season. It is important to note that the region receives an average of 5-10 AARs every year according to Chakraborty et al[9]. The IO region experiences an increase in the number of BC AARs by 3-5 AARs. The PC region receives less (<10) BC AARs in season19. The regional variability of the differences in the BC AARs can be explained by a large-scale transport pathway (not shown) of AARs. Panels B and C in Figure 1 show the land ice albedo over the Antarctic region measured by TERRA MODIS satellite during December 2018 and 2019. It appears that the land ice, which is further away from the Amazonia and is protected by the sea ice, is significantly darker in 2019 than in 2018. Owing to the absence of sea ice albedo data, this figure attempts to portray an idea of the snow and ice darkening differences during the 2019 wildfire season to the previous year.

Figure 2 shows the differences in the SIE loss between the two seasons. It appears that in season18, the SIE loss was higher over the IO and Ross regions. The difference is statistically significant

over the Ross region. The SIE loss is higher over the Weddell region during the BC days; however, not statistically significant. In season19, the SIE loss in Weddle significantly increased over the Weddell region with a loss of more than 33000 $Km^2$ of sea ice during the BC days. Very interesting pattern changes have been noticed during the BC days (when BC AARs are present over the region) and non-BC days. The Ross and IO regions experienced a higher loss in SIE during BC days, significantly over the Ross region. These results show the importance of the BC AARs on SIE loss over the regions that experienced higher BC AARs in 2019. The BA area is least affected and shows a reversal in the SIE loss with higher loss (although statistically insignificant) during season18, presumably due to higher precipitation in 2019[27]. The PC region appears to be least affected by the presence or absence of BC AARs as the number of BC AARs arriving over the PC region in season19 is less than that in season18 and is the farthest region from the Amazonia both for the westerly and well as easterly wind flows.

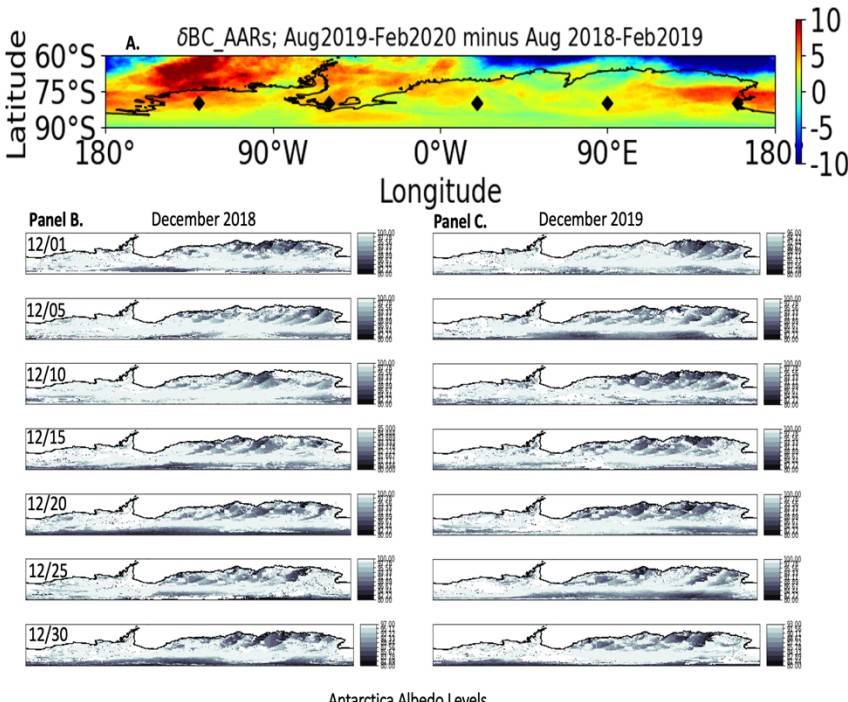

**Figure 1.** (A) Differences in the number of BC AAR occurrences over the Antarctic region during season19 and season18. Land albedo over the Antarctic region in December 2018 (panel B) and December 2019 (panel C).

The matrix profile analysis can address some of the loss patterns we have seen in Figure 2. The discord plot for the five Antarctic regions is depicted in Figure 3. The x-axis represents the number of days our study was conducted. Our study spanned three years, or roughly, 1078 days, with a few days missing. The y-axis represents the obtained matrix profile values for the relevant parameter, which is sea ice extent. The green line indicates the upper limit threshold we applied to obtain the discords. Values that surpass the green line are identified as discords. The discords (Table 2) or the extreme melt events are higher in the Ross region in season 19 (31) as compared to season 18 (26). Higher discords explain the higher SIE loss in season19 than season18. The Weddell region experiences the highest number of discords in both seasons - a total of 68 discords - in tandem with the highest SIE loss over that region than any other region. However, the number of discords over there cannot explain the higher SIE loss over the IO and Weddell regions in season19 than season18. Hence, we conduct RF, EL, and CD analyses to understand the feature importance and the causes behind such losses in SIE over the Weddell and IO regions.

The BA region had fewer discords than the other four regions. Even though the BA is located between the Ross and Weddell Seas, this did not appear to amplify the unusual behavior of sea ice extent. BC AARs and SIE loss over the BA region appears to be least connected because of the heavy precipitation from atmospheric rivers in 2019 that contributed to rapid increase in the snow height over the west Antarctic region[27]. The number of discords over the PC region is lower in season19 than season18 as well as the presence of BC AARs (Figure 1). We exclude the PC and BA regions from our RF and EN analyses.

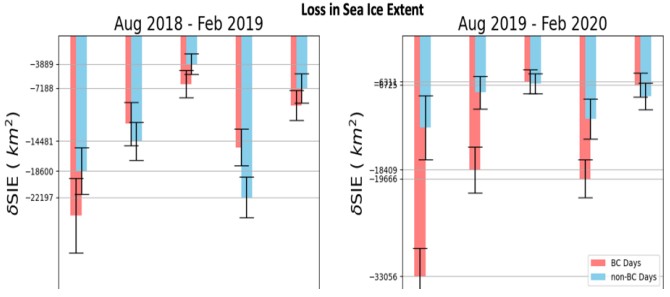

**Figure 2.** Mean and standard errors of SIE loss over different regions in season18 and season19 during BC and non-BC days.

Figure 4 shows that over the Weddell region, SWD or the incoming sunlight, SWU or the reflected sunlight (or a representation of the albedo or snow darkness), LWU (longwave upward or emitted radiative flux), LWD (longwave downward radiative flux), Temperature (T), and relative humidity (Q) are the major factors impacting the SIE loss over there. the regression coefficients also match with the RF analysis; however, SWU and LWD appear to have the largest coefficients. LWD is the LWU radiated back to the planet due to cloud cover and greenhouse gas. Owing to a strong dependency between LWU and T, the model penalizes the coefficient of T.

The PCMCI analysis aimed to unravel the intricate interactions governing these relationships, particularly concerning the sea ice extent as the parameter of interest. Table 3 summarizes the results from the causal discovery for three different lag times. In the Weddell Sea region, shortwave downward radiation (SWD) and longwave upward radiation (LWU) negatively influence sea ice extent (↓), while longwave upward radiation (LWU) negatively and shortwave upward radiation (SWU) positively impact sea ice extent (↑). Thus, albedo (or SWU) and temperature (or LWU and T) are two primary factors governing the SIE loss.

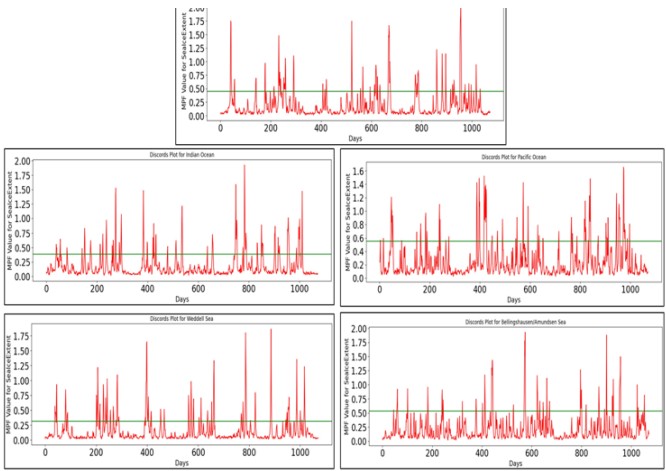

**Figure 3.** Discord Plot for Ross Sea, Indian Ocean, Pacific Ocean, Weddell Sea and Bellingshausen/Amundsen Sea.

Over the Ross region, the most important factors from RF analysis appear to be SWU, LWD, LWU, and Q. From the EL analysis, SWU, LWD, LWU, T and Q are the most important factors that govern the melting. In the Ross Sea, longwave upward radiation (LWU) demonstrates a positive influence on sea ice extent (↑) at all time lags, while both longwave downward radiation (LWD) and temperature (T) show negative effects (↓) at a time lag of 3. Similar feature importance is also observed over the IO region. From RF analysis, the most important features are SWU, LWD, T, and Q. EL analysis shows that the coefficients of SWU, LWD, and T are the highest among all the parameters. As Table 3 shows, in the Indian Ocean region, the results indicate that increased wind speed (V) and shortwave upward radiation (SWU) positively influences sea ice

extent (↑), while longwave downward radiation (LWD) negatively influences sea ice extent (↓) at all time lags. Furthermore, at time lags of 2 and 3, precipitation (PPT) exhibits negative cause (↓). From CD analysis, SWU, LWU, LWD, and T are directly causing SIE loss. CD analysis also shows that PPT (precipitation) and V (meridional wind) are also directly related to SIE. However, their importance is not captured in RF and EL analyses.

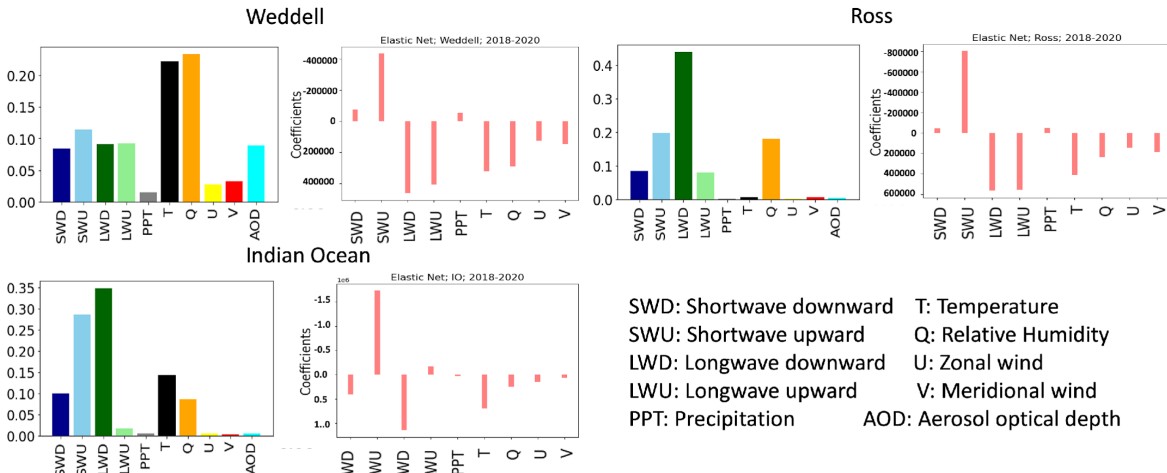

**Figure 4.** Feature importance from RF analysis and coefficients of the elastic net regression over Weddell, Ross, and IO regions.

| Regions | Time lag = 1 | Time lag = 2 | Time Lag = 3 |
|---------|--------------|--------------|--------------|
| IO | V↑, LWD ↓, SWD↓, SWU ↑ | V↑, LWD ↓, PPT↓, SWU ↑ | V↑, LWD↓, PPT↓, LWU↑, SWU ↑ |
| PC | SWD↓ | LWU↑, LWD ↓, SWU ↑ | V↑, LWD ↓, SWU ↑ |
| BA | V↑, PPT↑ | V↑, LWD↓ | V↑, LWD↓ |
| Ross | LWU↑, SWU↑ | LWU↑, LWD ↓ | LWU↑, LWD ↓, T↓ |
| Weddell | SWD↓, LWU↓ | LWU↓, SWU↑ | LWU↓, SWU↑ |

**Table 3.** Key features directly cause Sea Ice Extent in five different polar regions of the Antarctic basin. The symbol ↓ indicates negative inter-dependency and the symbol ↑ indicates positive inter-dependency strength.

For the Bel/Amundsen Sea, wind speed (V) is found to have a positive influence on sea ice extent (↑) at different time lags, while longwave downward radiation (LWD) exhibits a negative impact (↓) at time lags 2 and 3. Also, the results show precipitation (PPT) has a positive impact (↑) at time lag 1 on sea ice 28. Although RF and EL analyses are unable to capture that relationship, Table 3 indicates that SIE over the BA region depends on the large-scale features, cloud cover, and precipitation. In the Pacific Ocean region, the results show that longwave downward radiation (LWD) negatively (↓) and shortwave upward radiation (SWU) positively affect sea ice extent (↑) at time lag 2 and 3. These findings shed light on the specific features and their causal influences on sea ice extent in the different polar regions of the Antarctic basin, highlighting the complex dynamics involved. The symbols ↓ and ↑ in the table represent negative inter-dependency and

positive inter-dependency strength, respectively, providing a clear representation of the causal relationships uncovered by the analysis.

## 4. CONCLUSION AND DISCUSSION

We noticed that more than 10 BC AARs arrived over the Antarctic region in season19 from the Amazon wildfire region as compared to season18. Our results show that the Weddell region lost more sea ice than any other region in season19. SIE loss over the Ross, IO, and Weddell regions are higher and significantly different in season19 than season18 during the presence of BC AARs. The land ice was darker over the Antarctic peninsula in season19, which indicates lesser (higher) reflection (absorption) of the incoming sunlight. A higher amount of SWU or the solar energy reflected to space is very important that causes the SIE loss over the Weddell region along with LWU or the emitted longwave energy at the surface and temperature (T). SWU is also important over the Ross and IO regions - indicating a strong relationship between the albedo and SIE loss. A higher number of BC AARs are observed over the Ross, Weddell, and IO regions in season19, higher coefficients of SWU in the EL analysis, higher importance of SWU from the RF analysis, and a positive and direct relationship between SWU and SIE in the CD analysis suggest that the sea ice albedo as well as ice darkening by aerosols are very important for sea ice extent over these regions. Multiple experiments using satellite measurements to deduce the impact of Amazon wildfires on Antarctica SIE are unprecedented. Our study points out that in addition to global warming, the slash and burn of the Amazon rainforest can severely impact the SIE that protects the land ice. The relative influence of BC aerosols, global warming, and associated changes in other features on SIE needs to be explored to entangle their role in sea level rise. In the future, such practices will severely impact the already shrinking ice concentration, amplify the ice sheet melting both over the land and sea, and will amplify the sea level rise.

A matrix profile analysis was able to explain the role of extreme melt events on the higher SIE loss over the Ross region but was not able to explain the higher loss over the IO and Weddell regions during the BC days. The Matrix profile did capture the highest (lowest) discord over the Weddell (BA) region. It appears that the SIE loss over the IO and Weddell regions during BC days is not because of the extreme melt events as observed over the Ross region- rather from the continuous and steady melting. Our study was able to capture the relationships between the role of the presence of BC AARs, darkening of the snow and ice or albedo reduction, and the relationship between the reflected sunlight to the SIE loss over Weddell, Ross, and IO region using RF, CD, and EN analyses. However, RF and EN are unable to capture the importance of PPT and large-scale features over the BA region that CD analysis can deduce. Further analysis is needed to understand why the BA region has the least number of discords, how BC aerosols cause the higher SIE loss in IO and extreme SIE loss in the Weddell region during BC days in season19 than season18, and the continuous and steady melting of ice sheets is occurring over the Weddell and IO region. We will analyze the relationships between high discords and low discords on SIE to investigate the role of extreme and steady state melting processes, respectively. An in-depth analysis of how BC aerosols affect SIE is needed using long-term observational data. Owing to the importance of SWU on SIE, sea ice albedo needs to be included as another feature. Especially, the multi-domain relationships between the features and SIE can be explored to better understand the associated patterns in the SIE loss using multi domain anomalies from the gridded time series data. Since this study uses time series datasets, it is important to identify the neighborhoods that exhibit high amounts of snowmelt using spatio-temporal datasets to further explore how BC aerosols initiate the SIE melting and how the melting processes grow and expand spatially with time. These are beyond the scope of the current analysis and will be investigated in the future.

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
