# OpenReview forum: "Understanding the Role of 2019 Amazon Wildfires on Antarctic Sea Ice Extent Using Data Science Approaches"
_KDD.org/2023/Workshop/Fragile_Earth — KDD 2023 Workshop Fragile Earth Submission_

### Official Review · Reviewer_th8k · 2023-07-13
**Review of "Understanding the Role of 2019 Amazon Wildfires on Antarctic Ice Sheet Melting Using Data Science Approaches."**

**Rating:** 6
**Confidence:** 3

**Review:**

This paper runs a set of experiments to understand the impact of the wildfires on Antarctic ice cap. Being from outside this domain, I really appreciate the explanations made by the authors especially in the introduction section where how sea ice melts are affecting/accelerating the sea level increase as well as the black carbon aerosols effect on these. Also, it is well aligned with the workshop theme. That being said, I have a couple of questions/clarifications needed from the authors.

a. Three methods were applied on the data i.e. Random Forest, Elastic Net, and Matrix profiles. I understand that these were previously studied methods (references 17-21), but these are usually pretty basic models. Is it possible to put a discussion regarding what else can be done, what other approaches are more promising for such a task.

b. Table 2 datasets are not directly applicable. For example SMMR is daily averaged values in sqkm. whereas others are spatial (in degrees or sub-degrees). Do you replicate these for all cells? I am not sure I can follow how that is aggregated for the specific dataset.

c. Too many acronyms in Section 4 Results. Do you mind repeating their correspondence at the beginning.

d. Random Forest is not in the Section 4. I understand that it is used for learning about the feature importances. I, however, am not sure since the details were omitted. I think this part needs a clarification for sure.

e. Figure 4 is just too small to read, it needs to be clarified better.

Regarding the overall paper, I understand that the expectation is that the black carbon aerosols affect the ice cap. And we all know it in terms of physical effect (instead of an empirical analysis). Now, the paper tries to prove it with experiments. However, I want to understand in a clearer way, what is the effect? what should be done? how should we react to all this analysis. I think we may need a discussion section to make sense of the results.

---

### Official Review · Reviewer_LmWC · 2023-07-14
**Review for "Understanding the Role of 2019 Amazon Wildfires on Antarctic Ice Sheet Melting Using Data Science Approaches"**

**Rating:** 6
**Confidence:** 2

**Review:**

Summary : Using Machine learning methods such as Random Forest and Elastic Net regression effects on Black carbon on acceleration of Antarctic ice sheet melting has been provided.

Strengths :
- Problem is well explained as an important environmental challenge .
- Machine Learning methods have been applied .
-  Data has been explained sufficiently.

Weaknesses :
- Too many abbreviations have been used making readability challenging.
- Results while promising aren't presented well. It would also be nice to see comparison of other methods to study and explain ice sheet melting.
- Technical details about methods used are incomplete.
- Future work and direction and discussion about method limitations are lacking.

Questions : what is relative impact of Black Carbon vs other factors causing and accelerating ice cap melting ?

Limitations : Readability of paper needs to be worked on and other methods need to be tested.

---

### Official Review · Reviewer_SPuZ · 2023-07-16
**Review for "Understanding the Role of 2019 Amazon Wildfires on Antarctic Ice Sheet Melting Using Data Science Approaches"**

**Rating:** 7
**Confidence:** 4

**Review:**

Summary:

This paper investigates the impacts of the black carbon (BC) aerosols generated during the 2019 extreme Amazon wildfire events on the sea ice extent over the Antarctic region. In experiments, the authors employ various machine learning methods to understand the role of aerosols on the exacerbated sea ice loss in terms of their extent in km2 including Random Forest (RF), Elastic Net (EN), and Matrix profile analyses that are described in detail in the methodology section.

Strengths:
- The paper is easy to follow.
- Experiments are comprehensive.
- I appreciate the authors provide the standard deviations of SIE loss over different regions.

Weaknesses:
- Related work section is missing. I suggest the authors add a related work section and introduce the prior works.

---

### Decision · Program_Chairs · 2023-07-19

**Decision:**

Accept (Oral)

**Comment:**

Congratulations!

We are pleased to inform you that your submission: Understanding the Role of 2019 Amazon Wildfires on Antarctic Ice Sheet Melting Using Data Science Approaches has been accepted to The KDD 2023 Workshop Fragile Earth: AI for Climate Sustainability - from Wildfire Disaster Management to Public Health and Beyond.

Camera ready deadline is ** July 24 AOE **.  Please log in to OpenReview and prepare your camera-ready version based on the reviews. Formatting rules are the same as for the initial submission and submissions must adhere to KDD 2023 guidelines available at https://authors.acm.org/proceedings/production-information/taps-production-workflow.

Again, congratulations on the acceptance of your paper!  We look forward to seeing you at the workshop on Aug 7, 2023.

The Fragile Earth Workshop Proceeding Chairs